# Prevalence of *Toxoplasma gondii* in Wild American Mink (*Neogale vison*): The First Serological Study in Germany and Poland

**DOI:** 10.3390/pathogens13020153

**Published:** 2024-02-07

**Authors:** Mike Heddergott, Jutta Pikalo, Franz Müller, Natalia Osten-Sacken, Peter Steinbach

**Affiliations:** 1Department of Zoology, Musée National d’Histoire Naturelle, 2160 Luxembourg, Luxembourg; psteinbach@web.de; 2Institute of Parasitology, University of Veterinary Medicine Vienna, Veterinärplatz 1, 1210 Vienna, Austria; jutta.pikalo@vetmeduni.ac.at; 3Wildlife Biology Working Group, Justus-Liebig-University of Giessen, 35392 Giessen, Germany; franzgersfeld@web.de; 4Institute for Veterinary Sciences, Nicolaus Copernicus University, 87-100 Toruń, Poland; natalioss.osten@gmail.com; 5Faculty of Chemistry, Georg-August University of Göttingen, 37073 Göttingen, Germany

**Keywords:** *Toxoplasma gondii*, enzyme-linked immunosorbent assay (ELISA), American mink, *Neogale vison*, wildlife, invasive species, zoonosis

## Abstract

*Toxoplasma gondii* is an obligate intracellular protozoan that causes toxoplasmosis in warm-blooded animals. Although most infections in humans and animals are subclinical, an infection can nevertheless be fatal. One of the important characteristics in the epidemiology of this parasite is waterborne transmission. The American mink (*Neogale vison*), a mammal closely adapted to freshwater ecosystems, is a potential sentinel for *T. gondii*. We analysed meat juice from the heart of 194 wild minks collected between 2019 and 2022 in five study areas from Germany and Poland and tested for the presence of antibodies against *T. gondii*. The analysis was performed using a commercial enzyme-linked immunosorbent assay test (ELISA). Antibodies were detected in 45.36% (88/194, 95% confidence interval (CI): 38.39–52.41%) of the analysed animals. While the prevalence values ranged from 37.50% to 49.30%, there was no significant difference in seroprevalence between the study areas. Juveniles were less likely to carry *T. gondii* antibodies than adults (odds ratio: 0.216), whereas there was no significant difference in prevalence between the sexes (odds ratio: 0.933). The results of our study show that contact with *T. gondii* is widespread in minks, and the parasite is common in inland freshwater ecosystems in Germany and Poland. This indicates that watercourses play an important role in the spread of *T. gondii* oocysts.

## 1. Introduction

*Toxoplasma gondii* is an intracellular protozoan parasite with a global distribution. Potentially all warm-blooded vertebrates, including humans, can act as intermediate hosts [1]. The parasite has an indirect life cycle in which felids (family: Felidae) are the only definitive hosts. They excrete oocysts into the environment via faeces. Infection occurs horizontally through the ingestion of water or food contaminated with sporulated *T. gondii* oocysts or through the consumption of tissues from animals infected with encystised bradyzoites, as well as vertically through transplacental transmission of tachyzoites [1]. There is also the possibility of sexual transmission of toxoplasmosis [2]. Waterborne transmission of *T. gondii* is a growing problem. In the past, infections in humans have been linked to contaminated watercourses [3,4,5,6,7]. Sporulated oocysts are resistant to numerous environmental conditions and can remain infectious in freshwater for many months at low temperatures [1].

In addition to food, semi-aquatic mammals can also become infected via water contaminated with oocysts. The loss and degradation of wetlands due to intensive agriculture and urbanisation can facilitate the transport of oocysts from terrestrial to aquatic areas [8,9,10,11,12,13]. Furthermore, the presence and abundance of feral and domestic cats can increase the risk of *T. gondii* infection for wildlife species [14]. Mammals from a high tropical level in the aquatic ecosystem and can be good sentinel species for the detection of various pathogens, including *T. gondii* [15,16].

The American mink (*Neogale vison*; syns.: *Neovison vison*, *Mustela vison*), referred to henceforth as mink, is a Nearctic semi-aquatic representative of the Mustelidae family. While the species has its original distribution area in North America, it was introduced to Europe, Asia, and South America for fur farming. Following accidental escapes and deliberate releases [17], the mink has become established in Europe. It is geographically widespread in its invasive range, with the main distribution area being in northern and eastern Europe [18]. Minks were first observed in the wild in Germany in the 1950s [19] and in Poland in the 1960s [20]. The species’ main distribution area in Germany is in the eastern and northern [21] and south-western part of the country [22], whereas in Poland the mink is distributed across the entire country, with a focus on the northern lowlands [22,23,24]. The mink is one of the most invasive mammals, with major negative ecological and economic impacts [25]. As a semi-aquatic predator, the mink is an inhabitant of coasts, watercourses, lakes, ponds, and other reservoirs, including their neighboring habitats [26]. As omnivores, minks have a wide range of food that includes rodents, shrews, birds, amphibians, fish, and crustaceans. The diet composition of minks varies depending upon their habitat type. Mammals, fish, and amphibians are the most important food resources in rivers, whereas birds and fish predominate in the diet of minks living near lakes and ponds [27].

Serological studies on *T. gondii* infections in minks were mostly performed on animals from fur farms [28,29,30,31,32,33,34], while studies on wild minks are generally rare and often based on small sample sizes [35,36,37]. The aim of the present study was to analyse for the first time the seroprevalence of *T. gondii* in a larger sample of wild minks in order to assess the contamination of freshwater ecosystems in Germany and Poland.

## 2. Materials and Methods

### 2.1. Ethical Statement

The mink is an invasive species in both Germany and Poland. It is not protected by law in either country, and it can be hunted by licensed hunters [22,38]. All hunted animals were harvested legally and made available to the authors for this study. No animal was killed for the purpose of being used in this study.

### 2.2. Sample Collection

A total of 194 minks were collected as part of convenience sampling. All animals, which were shot during hunting activities (n = 177) in the study period between September 2019 and October 2021, as well as road-killed animals (n = 17), were included in this study. The carcasses were collected in three study areas in eastern Germany, Artern (AT: 51°20′ N/11°19′ E; size of the study area 50 km^2^), Havelland (HB: 52°49′ N/12°05′ E; 150 km^2^), and Torgau (TO: 51°35′ N/13°01′ E; 35 km^2^), and two study areas from western Poland, Pieńsk (PK: 51°15′ N/15°02′ E; 50 km^2^) and Słubice (SB: 52°22′ N/14°33′ E; 25 km^2^) (Figure 1).

The heart of each animal was removed and examined macroscopically. None of the hearts showed macroscopic lesions. They were then stored in a sealable polyvinyl bag at −20 °C for 48 to 72 h. The serological examination was performed on meat juice. As described by Nielsen et al. [39], each heart was placed in a plastic funnel (Mollenkopf GmbH, Stuttgart, Germany) for thawing at room temperature, and the drained meat juice was collected in tubes. The collected meat juice was clean and free of dirt, and it was refrozen at −20 °C.

As described by Heddergott et al. [40], the age of the mink was determined using the growth lines in the cementum of a fang from the upper jaw. Animals were categorised into two age classes, juvenile and adult, with adult individuals having one or more growth lines and juvenile individuals having none. The data set analysed for this study comprised 97 males and 97 females, with 83 adults and 111 juveniles.

### 2.3. Determination of Antibodies to T. gondii by ELISA

The meat juice was analysed for IgG antibodies against *T. gondii* using the ID Screen^®^ Toxoplasmosis Indirect Multi-species ELISA kit (bioMérieux, Lyon, France) according to the manufacturer’s instructions. The determination was carried out according to the latest manufacturer’s instructions, and it was performed with a sample volume of 50 µL at a dilution of 1:2. Although the antibody concentration in meat juice is lower than in serum, serological analyses showed that both media correlate well [41]. Meat juice has been successfully used in the past to monitor for antibodies against *T. gondii*, including in mammalian predators [34,42,43]. The optical density (OD) was measured at 450 nm using a microplate reader (Tecan Infinite^®^ F50 Plus, Tecan Trading AG, Männedorf, Switzerland). Using internal positive and negative controls, the ratio between the sample and the positive control (S/P ratio) was calculated using the following formula:SP%=ODvalueofthesample−ODvalueofthenegativecontrolODvalueofthepositivecontrol−ODvalueofthenegativecontrole×100

Samples showing S/P% ≤ 25% were classified as negative for the presence of *T. gondii* antibodies, those within the range of 25–30% as equivocal, and those with S/P% ≥ 30% as positive. All samples were tested twice and samples with equivocal results were tested a third time.

### 2.4. Statistical Analysis

The prevalence of antibodies against *T. gondii* was estimated from the ratio of positive samples to the total number of samples tested, with a 95% confidence interval (95% CI). Effects were considered statistically significant if *p* < 0.05. We used a *χ*^2^-test to test for differences in prevalence between study areas. We performed a logistic regression to test for the effect of sex and age category on the presence of *T. gondii* antibodies. The statistical analyses were performed in program SPSS v.22 (SPSS Inc., Chicago, IL, USA).

## 3. Results

We identified the presence of *T. gondii* antibodies in 88 of the 194 (45.36%; 95% CI 38.39–52.41%) analysed minks. None of the meat juice samples tested had a dubious result. Seropositive minks were found in all five study areas (Table 1). Although there were major differences between the seroprevalence of some study areas, including Pieńsk (37.50%) and Havelland (49.30%), there was no significant overall difference between the study areas (*χ*^2^ = 1.06; df = 4; *p* = 0.901). While sex had no influence on the presence of *T. gondii* antibodies, there was a significant difference between the age groups (*p* ≤ 0.001). Adults (prevalence: 66.27%; 95% CI 56.10–76.52%) were more likely to be seropositive than juveniles (prevalence: 23.73%; 95% CI 17.63–41.86%; Table 2).

## 4. Discussion

In the original distribution area, the seroprevalence of *T. gondii* in wild minks was between 40 and 100% [13,44,45,46]. In areas where the species has been introduced, including South America, Europe, and Asia, a seroprevalence of between 26 and 100% has been reported [34,47,48,49,50,51]. However, seroprevalence values were mostly based on small sample sizes and sometimes on individual animals. To our knowledge, the present study is the first serological study on wild minks in Germany. There is only one report from eastern Poland, in which a single wild mink was tested using an ELISA and the direct agglutination test (DAT), and it was shown to be seropositive in both tests [52].

Serological studies on wild mink populations with comparable sample sizes (≥200 individuals) were conducted in Denmark and Spain. The seroprevalence of 45.36% (95% CI: 38.39–52.41%) determined in the present study was comparable to the prevalence of 53.6% (126/235; 95% CI: 40.0–60.0%) reported from Denmark [34], which was also determined using ELISA based on meat juice. A significantly higher seroprevalence was reported from Spain [51], where *T. gondii* antibodies were found in 78.8% (534/678; 95% CI: 75.5–81.8%) of wild minks using an indirect modified agglutination test (MAT) based on serum samples. Experimental studies have shown that ELISA and MAT provide comparable results in terms of both sensitivity and reaction speed [53,54].

Studies on wild minks show significant differences in seroprevalence between study regions in South America [47,50] and in Europe [34,51]. In wild minks from Chile, the differences between study areas were associated with locally higher cat density [47,50]. Wild minks on Danish islands (Bornholm and Zealand) were significantly more seropositive than animals from the mainland, which the authors associate with regional differences in *T. gondii* infection of prey [34]. Different ecological and climatic characteristics as well as the higher density of the final hosts are cited as reasons for the significant differences in seroprevalence between bioregions in northern Spain [51]. In contrast, results of other studies from Argentina [49] and the USA [13] showed no differences in the seroprevalence of wild minks between different study areas, which is consistent with the results of our findings.

In general, several studies have reported a positive relationship between high *T. gondii* seroprevalence in wildlife and increased cat density [8,47,50,51,55]. To our knowledge, there are no current estimates of cat densities in the areas included in this study. According to information from regional hunters, all study areas harboured a fairly high density of domestic cats and feral domestic cats, which could explain the lack of differences in seroprevalence between the study areas. This assumption is supported by the fact that no regional differences in seroprevalence have been reported in omnivores, such as the raccoon (*Procyon lotor*), or in terrestrial predatory mammals, such as the red fox (*Vulpes vulpes*) in Germany [56,57].

Whether diet has an influence on the results we obtained cannot be conclusively assessed, as only a few and mainly regional studies on the feeding ecology of wild minks have been conducted in both Germany and Poland, with mostly small sample sizes [58,59].

As there was increased contact at older ages, the results suggest that the main route of infection was horizontal transmission in the minks studied, and thus it may be the consumption of prey with tissue cysts as well as the ingestion of oocysts from the environment [58,59]. However, depending on the season, these studies show a high proportion of aquatic prey, such as fish, as food. According to various reports, fish do not develop tissue cysts, and, in general, the involvement of fish in the epidemiology of *T. gondii* is still unclear. However, it is known that filter-feeding fish as well as mussels can ingest sporulated *T. gondii* oocysts from seawater and maintain their infectivity [60]. Furthermore, the ingestion of sporulated oocysts present in the water is possible. The high seroprevalence found in the predominantly piscivorous minks indicates a high fecal contamination of the freshwater ecosystem with oocysts. The few nutritional ecology studies from Germany and Poland [58,59,61] show that minks are opportunistic predators and have a wide range of prey. Seasonally, mammals and birds also play an important role in the diet of wild minks, which can then become infected with *T. gondii* tissue cysts by swallowing prey.

The age of the host is often identified as increasing the probability of a *T. gondii* infection, as the probability of contact with the parasite increases with age [62]. Our results showed a significant difference in seroprevalence between the age groups (juveniles and adults). Other studies on wild minks from Argentina [49], Chile [47], and Spain [51] came to the same conclusion, according to which older individuals were significantly more seropositive than younger individuals. In contrast, no significant difference was found between the age groups in another study on wild minks from Chile [50], although the authors point out a tendency towards higher prevalence in older animals.

Our results show no link between the presence of *T. gondii* antibodies and sex of the mink. This is consistent with results from studies on wild minks in Argentina [49] and Chile [47,50]. In contrast, a study on wild minks from Denmark found a significantly higher probability of the presence of *T. gondii* antibodies in males [34]. These authors hypothesised that the differences in seroprevalence between the sexes could have been due to different food preferences.

Surveillance of wild mink populations is appropriate because of conservation concerns for the Eurasian otter (*Lutra luta*) in our study areas and the highly endangered European mink *(Mustela lutrealo*) elsewhere. The mink potentially represents a useful sentinel species for monitoring waterborne pathogens, such as *T. gondii*. Toxoplasmosis has been reported in wild minks [63]. There is genetic evidence of *T. gondii* infection in wild minks in Poland [64], but no deaths associated with the protozoan have been confirmed in either Germany or Poland. Earlier studies on farmed minks showed that *T. gondii* infection in the species is mostly asymptomatic [65], whereas outbreaks, clinical cases, and spontaneous abortions have also been reported [28,29,66,67].

Demographic change and anthropogenic activity are altering the emergence and spread patterns of various pathogens [68]. As human populations, including domestic animals, increase, faecal contamination and its impact on human, domestic animal, and wildlife health will increase [69,70]. The results of our study once again support the growing evidence that freshwater ecosystems play an important role in the transmission of *T. gondii*. We cannot rule out the possibility that our relatively small sample does not reflect the active prevalence. Surveillance control studies, based on larger samples, are warranted and necessary to further clarify the occurrence, prevalence, and epidemiology of *T. gondii* in freshwater ecosystems and to assess its public health significance.

## 5. Conclusions

We analysed the sero-epidemiology of *T. gondii* infection in wild minks from different populations in Germany and Poland. Antibodies against *T. gondii* were present in animals from all study areas and all age groups. The high seroprevalence indicates a high degree of circulation of the parasite in the semi-aquatic habitat. Although almost half of all minks were *T. gondii* seropositive, the zoonotic risk was considered to be low if general hygiene guidelines were observed, and also because minks are not eaten.

## Figures and Tables

**Figure 1 pathogens-13-00153-f001:**
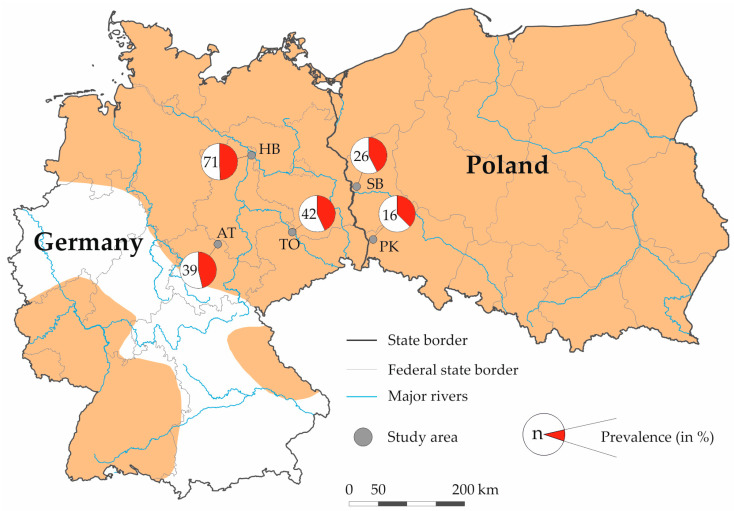
Sampling areas, sample size (*n*), and seroprevalence of *Toxoplasma gondii* in wild American minks (*Neogale vison*) from Germany and Poland. The orange area represents the geographic distribution of the American mink in Germany and Poland (after Vada et al. [22]). Germany: AT Artern (federal state: Thuringia), HB Havelland (Saxony-Anhalt), and TO Torgau (Saxony); Poland: SB Słubice (Lubusz) and PK Pieńsk (Lower Silesian).

**Table 1 pathogens-13-00153-t001:** Seroprevalence of *Toxoplasma gondii* in wild American minks (*Neogale vison*) from Germany and Poland.

State	Study Area	No. Tested	No. Positive	Prevalence in % (95% CI) ^1^
Germany	Artern	39	18	46.15 (30.54–61.86)
	Torgau	42	18	42.86 (27.93–57.87)
	Havelland	71	35	49.30 (37.67–60.93)
Poland	Pieńsk	16	6	37.50 (13.78–61.22)
	Słubice	26	11	42.31 (23.31–61.29)
Total		194	88	45.36 (38.39–52.41)

^1^ CI–confidence interval.

**Table 2 pathogens-13-00153-t002:** Results of logistic regression identifying predicators for the seroprevalence of *Toxoplasma gondii* in wild American minks (*Neogale vison*). The lines with no data indicate the reference category.

Coefficients	Estimate	s.e.	Odds Ration	95% CI ^1^ Odds Ration	z-Value	*p*-Value
(Intercept)	0.707	0.273	2.028	1.199–3.516	2.589	0.010
Sex—female	-	-			-	-
Sex—male	−0.069	0.310	0.933	0.508–1.718	−0.224	0.823
Age—adult	-	-			-	-
Age—juvenile	−1.531	0.312	0.216	0.116–0.395	−4.904	<0.001

^1^ CI–confidence interval.

## Data Availability

Not applicable.

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
