# Peer review of "Prevalence of Toxoplasma gondii in Wild American Mink (Neogale vison): The First Serological Study in Germany and Poland"

_pathogens, 2024, doi:10.3390/pathogens13020153_

Round 1

Reviewer 1 Report

Comments and Suggestions for Authors

The research aimed to conduct a seroepidemiological survey of toxoplasmosis in the wild population of American Mink, as well as to establish possible risk factors (age, sex, and geographic area). Despite the simplicity of this epidemiological study, it holds significance for wildlife conservation guidelines and the prevention of zoonoses within the context of One Health. Minks are susceptible to viral agents, such as the one causing COVID-19, and understanding the health status of this wild population may be of interest to the international scientific community.

I suggest some adjustments to improve the manuscript:

Title: Replace "a first" with "the first."

Abstract: Include the odds ratios (OR) for the mentioned risk factors. Additionally, the conclusion is inadequate. The authors did not assess the detection of T. gondii oocysts in aquatic environments. Provide a relevant conclusion based on the study results.

Introduction: Include the dietary and ecological habits of the species. How could they have been infected? As predators, could they ingest tissue cysts from small animals? Or does infection only occur through the ingestion of oocysts in water and vegetables? This needs to be clarified for a better understanding of the study. Also, include this line of reasoning in the discussion.

Line 42: Disagree with the authors' statement that toxoplasmosis transmission through contaminated water only occurred in the past. The largest reported toxoplasmosis outbreak in Brazil recently happened due to contaminated water. Several articles are available on this topic (10.1111/tbed.13741 and 10.1007/s00436-023-07808-4).

Introduction must also contain information about the possibility of sexual transmission of toxoplasmosis (10.1093/aje/kwaa198, 10.1016/j.smallrumres.2013.08.008, 10.1016/j.jinf.2022.10.034 )

In the methodology, request the journal editors to confirm if the research ethics comply with the required standards.

Methods and Materials should contain information about the sampling. Was it convenience sampling? Was there a calculation of the sampled population to assess its representativeness?

Methods and Materials: Figure 1. Suggest removing prevalence values as they are results, and keeping only the number of samples from each region. The results are already presented in Table 1.

Methods and Materials: It is necessary to include the sensitivity and specificity values of the serological test used.

Methods and Materials: The significance level used in tests should be mentioned in the statistical analysis section, as well as the method of calculating 95% CIs (and referenced).

Lines 145-146: Clarify if this is a footnote.

Author Response

The research aimed to conduct a seroepidemiological survey of toxoplasmosis in the wild population of American Mink, as well as to establish possible risk factors (age, sex, and geographic area). Despite the simplicity of this epidemiological study, it holds significance for wildlife conservation guidelines and the prevention of zoonoses within the context of One Health. Minks are susceptible to viral agents, such as the one causing COVID-19, and understanding the health status of this wild population may be of interest to the international scientific community.

We would like to thank the reviewer for his thorough review of the manuscript and his comments. We have made some changes.

I suggest some adjustments to improve the manuscript:

Title: Replace "a first" with "the first."

>> We have changed the title.

Abstract: Include the odds ratios (OR) for the mentioned risk factors. Additionally, the conclusion is inadequate. The authors did not assess the detection of T. gondii oocysts in aquatic environments. Provide a relevant conclusion based on the study results.

>> We have changed the last sentence and added another sentence: “The results of our study show that contact with T. gondii is widespread in mink and the parasite is common in inland freshwater ecosystems in Germany and Poland. This indicates that watercourses play an important role in the spread of T. gondii oocysts.”

>> We have changed one sentence in the abstract and added the OR: “Juveniles were less likely to carry T. gondii antibodies than adults (odds ratio: 0.216), whereas there was no significant difference in prevalence between the sexes (odds ratio: 0.933).”

Introduction: Include the dietary and ecological habits of the species. How could they have been infected? As predators, could they ingest tissue cysts from small animals? Or does infection only occur through the ingestion of oocysts in water and vegetables? This needs to be clarified for a better understanding of the study.

>> We have included two sentences on the ecology of the mink in the introduction: „As a semi-aquatic predator, the mink is an inhabitant of coasts, watercourses, lakes, ponds and other reservoirs, including their neighboring habitats []. As omnivores, minks have a wide range of food and include rodents, shrews, birds, amphibians, fish, and crustaceans. The diet composition of mink varies depending upon habitat type. Mammals, fish, and amphibians are most important food resources on rivers, whereas birds and fish predominate in the diet of mink living near lakes and ponds [].“

Also, include this line of reasoning in the discussion.

>> We have added a new section to the discussion: “The results suggest, as there was increased contact at older ages, that the main route of infection was horizontal transmission in the minks studied and thus it may be the consumption of prey with tissue cysts as well as the ingestion of oocysts from the environment. In both countries, Germany and Poland, only a few studies have been carried out in the past on the feeding ecology of wild mink, mainly regional studies with mostly small sample sizes [49,50]. However, depending on the season, these studies show a high proportion of aquatic prey such as fish as food. According to various reports, fish do not develop tissue cysts and in general the involvement of fish in the epidemiology of T. gondii is still unclear. However, it is known that filter-feeding fish as well as mussels can ingest sporulated T. gondii oocysts from seawater and maintain their infectivity []. Furthermore, the ingestion of sporulated oocysts present in the water is possible. The high seroprevalence found in the predominantly piscivorous minks indicates a high fecal contamination of the freshwater ecosystem with oocysts. The few nutritional ecology studies from Germany and Poland [49,50,] show that minks are opportunistic predators and have a wide range of prey. Seasonally, mammals and birds also play an important role in the diet of wild mink, which can then become infected with T. gondii tissue cysts by swallowing prey.”   

Line 42: Disagree with the authors' statement that toxoplasmosis transmission through contaminated water only occurred in the past. The largest reported toxoplasmosis outbreak in Brazil recently happened due to contaminated water. Several articles are available on this topic (10.1111/tbed.13741 and 10.1007/s00436-023-07808-4).

>> Thank you for pointing this out. We have changed the sentence to: “Infections in humans have been linked to contaminated watercourses [].”

Introduction must also contain information about the possibility of sexual transmission of toxoplasmosis (10.1093/aje/kwaa198, 10.1016/j.smallrumres.2013.08.008, 10.1016/j.jinf.2022.10.034)

>> We have taken this into account and added a sentence: “There is also the possibility of sexual transmission of toxoplasmosis [].”

In the methodology, request the journal editors to confirm if the research ethics comply with the required standards.

>> Thank you for pointing this out. We think our "Ethical Statement" is sufficient for our study.

Methods and Materials should contain information about the sampling. Was it convenience sampling? Was there a calculation of the sampled population to assess its representativeness?

>> Information on the sample can be found under Material and method (Sample collection, last section). We have changed and adapted the sentence to make it easier to understand: “The data set analyzed for this study comprised 97 males and 97 females, 83 adults and 111 juveniles.”

>> The sample population was not calculated to assess representativeness. Except for one study area, hunting of the mink is sporadic in all other areas. An objective assessment is therefore not possible. All included animals in this study were shot during the hunting and study period. We have therefore changed and added an additional sentence under the section „Sample collection“: A total of 194 mink were collected. “All animals which were shot during hunting activities (n = 177) in the study period between September 2019 and October 2021 as well as road-killed animals (n = 17) were included in the study.”  

Methods and Materials: Figure 1. Suggest removing prevalence values as they are results, and keeping only the number of samples from each region. The results are already presented in Table 1.

>> Due to various discussions in the past regarding the presentation of the map, we would like to retain the map in its current form. Maps in this form, showing the location of the study areas, sample size and prevalence, have proved their worth in the past. 

Methods and Materials: It is necessary to include the sensitivity and specificity values of the serological test used.

>> Thank you for this note. We have made changes to one part of the text: “The determination was carried out according to the latest manufacturer's instructions and was performed with a sample volume of 50 µl at a dilution of 1:2. Although the antibody concentration in meat juice is lower than in serum, serological analyses showed that both media correlate well []. Meat juice has been successfully used in the past to monitor for antibodies against T. gondii, including in mammalian predators [].” 

section, as well as the method of calculating 95% CIs (and referenced).

>> We have added another sentence: “Effects were considered statistically significant if p < 0.05.”

Lines 145-146: Clarify if this is a footnote.

>> Due to the new Table 2, we have removed the footnote.  

Reviewer 2 Report

Comments and Suggestions for Authors

Dear authors,

the manuscript titled "Prevalence of Toxoplasma gondii in wild American mink (Neogale vision): a first serological study in Germany and Poland" presents interenting data about Toxoplasma in wild fauna, with the zoonotic risk that it may introduce.

I find the manuscript clearly written and I think it could be published as it is. However, let me introduce some suggestions:

1. Table 2. Although I can calculate OR and CI95 with the date included in the table, I think it better to add in the proper table OR  and CI. I remember that in SPSS (for the las decade I've used R) you have the option similar to ExpB to optain OR and CI. If not you can calculate it taking into account that the OR is the exponential of the estamate, and CI can be calculated with exponential of estimate +/-1.96 s.e. OR explains in a way a measure of the influence of the factor/category.

In logistic tables I also prefer to indicate reference category in a first line with no data (-) and then include a second line with the other/s category/ies. It is easier to understand the referring category.

2. Conclusions: I think that a sentence about the possible zoonotic risk should be included.

3. Introduction and references: In this paper there are a lot of references (it is not a criticism). I think that in some of the indications you include too much. For instance, line 66, 23-29. I think you can eliminate 27, as it is related to Asis, and this paper is referred to Europe. In general, I prefer to include the first and inportant one, one or two in the middle related directly with my research and one very near in time to my research, trying to maintain three or four references (only as a suggestion).

4. As a comment, 194 is a low number as a sampling size, and that low number may cause no differences to be found between areas or in other factors (low power in analyses; high beta error). Of course, I know that it is very difficult to obtain better number in wild animals.

Best regards,

Author Response

Dear authors,

the manuscript titled "Prevalence of Toxoplasma gondii in wild American mink (Neogale vision): a first serological study in Germany and Poland" presents interenting data about Toxoplasma in wild fauna, with the zoonotic risk that it may introduce.

We would like to thank the reviewer for the positive assessment of our manuscript. We have taken all comments into account.

I find the manuscript clearly written and I think it could be published as it is. However, let me introduce some suggestions:

  1. Table 2. Although I can calculate OR and CI95 with the date included in the table, I think it better to add in the proper table OR and CI. I remember that in SPSS (for the las decade I've used R) you have the option similar to ExpB to optain OR and CI. If not you can calculate it taking into account that the OR is the exponential of the estamate, and CI can be calculated with exponential of estimate +/-1.96 s.e. OR explains in a way a measure of the influence of the factor/category.

In logistic tables I also prefer to indicate reference category in a first line with no data (-) and then include a second line with the other/s category/ies. It is easier to understand the referring category.

>> We have changed Table 2 and added the calculation of the OR and CI as a column.

  1. Conclusions: I think that a sentence about the possible zoonotic risk should be included.

>> We have added one more sentence: “Although almost half of all mink were T. gondii seropositive, the zoonotic risk was considered to be low if general hygiene guidelines were observed, also because minks are not eaten. “

  1. Introduction and references: In this paper there are a lot of references (it is not a criticism). I think that in some of the indications you include too much. For instance, line 66, 23-29. I think you can eliminate 27, as it is related to Asis, and this paper is referred to Europe. In general, I prefer to include the first and inportant one, one or two in the middle related directly with my research and one very near in time to my research, trying to maintain three or four references (only as a suggestion).

>> We would keep the references as they are.

  1. As a comment, 194 is a low number as a sampling size, and that low number may cause no differences to be found between areas or in other factors (low power in analyses; high beta error). Of course, I know that it is very difficult to obtain better number in wild animals.

>> We agree that 194 animals tested in this study is a small number. We pointed this out in the last section of the discussion.

Reviewer 3 Report

Comments and Suggestions for Authors

The manuscript “Prevalence of Toxoplasma gondii in wild American mink (Neogale vision): a first serological study in Germany and Poland” By Heddergott et al., analyzed meat juice from the heart of 194 wild mink collected between 2019 and 2022 in five study areas from Germany and Poland for the presence of antibodies against T. gondii by commercial ELISA. The study indirectly assessed the contamination of freshwater ecosystems in Germany and Poland by T. gondii and found that contact with T. gondii is widespread in American mink and that T. gondii oocysts are common in freshwater ecosystems in Germany and Poland, as has been previously found in other countries, including Europe.

The manuscript is well written, and the conclusions are related to the results.

As main comments, additional details would be helpful about the access and collection of animal tissues (hearts) and other samples from the American minks. Were tissues observed to be in good health? How dirty/clean were the meat juices collected?

The authors only analyzed serologically the drained meat juice from the hearts after thawing them. Did they authors attempt to collect serum samples from the same animals? Since heart tissues were collected, was DNA extraction from the heart tissue not possible, or are the author planning to extract DNA in the future? Detection of antibodies only indicates that the animals had contact with the parasite, but detection of DNA by specific PCR detection of the parasite will confirm the presence of the parasite in the animals.

Please check the scientific name “Neogale vison” (line 53) or “vision” in title and abstract and correct.

Minor comments

In the tittle there is an unneeded extra space/separation after “and” and “Poland”

Author Response

The manuscript “Prevalence of Toxoplasma gondii in wild American mink (Neogale vision): a first serological study in Germany and Poland” By Heddergott et al., analyzed meat juice from the heart of 194 wild mink collected between 2019 and 2022 in five study areas from Germany and Poland for the presence of antibodies against T. gondii by commercial ELISA. The study indirectly assessed the contamination of freshwater ecosystems in Germany and Poland by T. gondii and found that contact with T. gondii is widespread in American mink and that T. gondii oocysts are common in freshwater ecosystems in Germany and Poland, as has been previously found in other countries, including Europe.

The manuscript is well written, and the conclusions are related to the results.

We would like to thank the reviewer for their comments.

As main comments, additional details would be helpful about the access and collection of animal tissues (hearts) and other samples from the American minks. Were tissues observed to be in good health? How dirty/clean were the meat juices collected?

>> We have added a sentence in the section "Sample collection": „ The heart of each animal was removed and examined macroscopically. None of the hearts showed macroscopic lesions and they were then stored in a sealable polyvinyl bag at - 20°C for 48 to 72 hours.” We have changed another sentence: „The collected meat juice was clean and free of dirt and was frozen at -20°C."

The authors only analyzed serologically the drained meat juice from the hearts after thawing them. Did they authors attempt to collect serum samples from the same animals? Since heart tissues were collected, was DNA extraction from the heart tissue not possible, or are the author planning to extract DNA in the future? Detection of antibodies only indicates that the animals had contact with the parasite, but detection of DNA by specific PCR detection of the parasite will confirm the presence of the parasite in the animals.

>> We are aware that a serological study only indicates the contact of the host with the parasites. As the hearts were collected by the hunters on site, it was not possible for us to collect blood samples and obtain serum samples. Due to a lack of financial resources, it was not possible to carry out genetic analyses to detect the parasite.

Please check the scientific name “Neogale vison” (line 53) or “vision” in title and abstract and correct.

>> It was changed.

Minor comments

In the tittle there is an unneeded extra space/separation after “and” and “Poland”

>> It was changed.

Reviewer 4 Report

Comments and Suggestions for Authors

Mike Heddergott and co-authors analyzed the sero-epidemiology of T. gondii infection in wild mink from different regions in Germany and Poland. Antibodies against T. gondii were present in animals from all study areas and all age groups. The high seroprevalence indicates a high degree of circulation of the parasite in the semi-aquatic habitat.

The authors used ELISA to determine seropositivity. It is unclear how many samples were positive (S/P% ≥ 30%) and how many intermediate (S/P% =25-30%). It is necessary to clarify.

Author Response

Mike Heddergott and co-authors analyzed the sero-epidemiology of T. gondii infection in wild mink from different regions in Germany and Poland. Antibodies against T. gondii were present in animals from all study areas and all age groups. The high seroprevalence indicates a high degree of circulation of the parasite in the semi-aquatic habitat.

The authors used ELISA to determine seropositivity. It is unclear how many samples were positive (S/P% ≥ 30%) and how many intermediate (S/P% =25-30%). It is necessary to clarify.

>> Thanks for the hint. We have added a sentence to the results: “None of the meat juice samples tested had a dubious result.”